# FUNDAMENTAL LIMITS AND TRADEOFFS IN INVARIANT REPRESENTATION LEARNING

## ABSTRACT

Many machine learning applications involve learning representations that achieve two competing goals: To maximize information or accuracy with respect to a target while simultaneously maximizing invariance or independence with respect to a subset of features. Typical examples include privacy-preserving learning, domain adaptation, and algorithmic fairness, just to name a few. In fact, all of the above problems admit a common minimax game-theoretic formulation, whose equilibrium represents a fundamental tradeoff between accuracy and invariance. In this paper, we provide an information theoretic analysis of this general and important problem under both classification and regression settings. In both cases, we analyze the inherent tradeoffs between accuracy and invariance by providing a geometric characterization of the feasible region in the information plane, where we connect the geometric properties of this feasible region to the fundamental limitations of the tradeoff problem. In the regression setting, we also derive a tight lower bound on the Lagrangian objective that quantifies the tradeoff between accuracy and invariance. Our results shed new light on this fundamental problem by providing insights on the interplay between accuracy and invariance. These results deepen our understanding of this fundamental problem and may be useful in guiding the design of adversarial representation learning algorithms.

## 1 INTRODUCTION

One of the fundamental tasks in both supervised and unsupervised learning is to learn proper representations of data for various downstream tasks. Due to the recent advances in deep learning, there has been a surge of interest in learning so-called invariant representations. Roughly speaking, the underlying problem of invariant representation learning is to find a feature transformation of the data that balances two goals simultaneously. First, the features should preserve enough information with respect to the target task of interest, e.g., good predictive accuracy. On the other hand, the representations should be invariant to the change of a pre-defined attribute, e.g., in visual perceptions the representations should be invariant to the change of perspective or lighting conditions, etc. Clearly, in general there is often a tension between these two competing goals of error minimization and invariance maximization. Understanding the fundamental limits and tradeoffs therein remains an important open problem.

In practice, the problem of learning invariant representations is often formulated as solving a minimax sequential game between two agents, a feature encoder and an adversary. Under this framework, the goal of the feature encoder is to learn representations that could confuse a worst-case adversary in discriminating the pre-defined attribute. Meanwhile, the representations given by the feature encoder should be amenable for a follow-up predictor of target task. In this paper, we consider the situation where both the adversary and the predictor have infinity capacity, so that the tradeoff between accuracy and invariance solely depends on the representations given by the feature encoder. In particular, our results shed light on the *best possible* tradeoff attainable by any algorithm. This leads to a Lagrangian objective with a tradeoff parameter between these two competing goals, and we study the fundamental limitations of this tradeoff by analyzing the extremal values of this Lagrangian in both classification and regression settings. Our results shed new light on the fundamental tradeoff between accuracy and invariance, and give a crisp characterization of how the dependence between the target task and the pre-defined attribute affects the limits of representation learning.

**Contributions** We geometrically characterize the tradeoff between accuracy and invariance via the information plane (Shwartz-Ziv & Tishby, 2017) analysis under both classification and regression settings, where each feature transformation correspond to a point on the information plane. For the classification setting, we provide a fundamental characterization of the feasible region in the information plane, including its boundedness, convexity, and extremal vertices. For the regression setting, we provide an analogous characterization of the feasible region by replacing mutual information with conditional variances. Finally, in the regression setting, we prove a tight information-theoretic lower bound on a Lagrangian objective that trades off accuracy and invariance. The proof relies on an interesting SDP relaxation, which may be of independent interest.

**Related Work** There are abundant applications of learning invariant representations in various downstream tasks, including domain adaptation (Ben-David et al., 2007; 2010; Ganin et al., 2016; Zhao et al., 2018), algorithmic fairness (Edwards & Storkey, 2015; Zemel et al., 2013; Zhang et al., 2018; Zhao et al., 2019b), privacy-preserving learning (Hamm, 2015; 2017; Coavoux et al., 2018; Xiao et al., 2019), invariant visual representations (Quiroga et al., 2005; Gens & Domingos, 2014; Bouvrie et al., 2009; Mallat, 2012; Anselmi et al., 2016), and causal inference (Johansson et al., 2016; Shalit et al., 2017; Johansson et al., 2020), just to name a few. To the best of our knowledge, no previous work studies the particular tradeoff problem in this paper. Closest to our work are results in domain adaptation (Zhao et al., 2019a) and algorithmic fairness (Menon & Williamson, 2018; Zhao & Gordon, 2019), showing a lower bound on the classification accuracy on two groups, e.g., source vs. target in domain adaptation and majority vs. minority in algorithmic fairness. Compared to these previous results, our work directly characterizes the tradeoff between accuracy and invariance using information-theoretic concepts in both classification and regression settings. Furthermore, we also give an approximation to the Pareto frontier between accuracy and invariance in both cases.

## 2 BACKGROUND AND PRELIMINARIES

**Notation** We adopt the usual setup given $(X, Y) \in \mathcal{X} \times \mathcal{Y}$, where $Y$ is the response, $X \in \mathbb{R}^p$ represents the input vector, and we seek a classification/regression function $f(X)$ that minimizes $\mathbb{E}\ell(f(X), Y)$ where $\ell : \mathcal{Y} \times \mathcal{Y} \to \mathbb{R}$ is some loss function depending on the context of the underlying problem. In this paper, we consider two typical choices of $\ell$: (1) the cross entropy loss, i.e. $\ell(y, y') = -y \log(y') - (1-y) \log(1-y')$, which is typically used when $Y$ is a discrete variable in classification; (2) the squared loss, i.e. $\ell(y, y') = (y - y')^2$, which is suitable for $Y$ continuous, as in regression. Throughout the paper, we will assume that all random variables have finite second-order moments.

**Problem Setup** Apart from the input/output pairs, in our setting there is a third variable $A$, which corresponds to a variable that a predictor should be invariant to. Depending on the particular application, $A$ could correspond to potential protected attributes in algorithmic fairness, e.g., the ethnicity or gender of an individual; or $A$ could be the identity of domain index in domain adaptation, etc. In general, we assume that there is a joint distribution $\mathcal{D}$ over the triple $(X, A, Y)$, from which our observational data are sampled from. Upon receiving the data, the goal of the learner has two folds. On one hand, the learner aims to accurately predict the target $Y$. On the other hand, it also tries to be insensitive to variation in $A$. To achieve this dual goal, one standard approach in the literature (Zemel et al., 2013; Edwards & Storkey, 2015; Hamm, 2015; Ganin et al., 2016; Zhao et al., 2018) is through the lens of representation learning. Specifically, let $Z = g(X)$ where $g(\cdot)$ is a (possibly randomized) transformation function that takes $X$ as input and gives the corresponding feature encoding $Z$. The hope is that, by learning the transformation function $g(\cdot)$, $Z$ contains as much information as possible about the target $Y$ while at the same time filtering out information related to $A$. This problem is often phrased as an adversarial game:

$$\min_{f, g} \max_{f'} \quad \mathbb{E}_{\mathcal{D}}[\ell(f \circ g(X), Y)] - \lambda \cdot \mathbb{E}_{\mathcal{D}}[\ell(f' \circ g(X), A)], \tag{1}$$

where the two competing agents are the feature transformation $g$ and the adversary $f'$, and $\lambda > 0$ is a tradeoff hyperparameter between the task variable $Y$ and the attribute $A$. For example, the adversary $f'$ could be understood as a domain discriminator in applications related to domain adaptation, or an auditor of sensitive attribute in algorithmic fairness. In the above minimax game, the first term corresponds to the accuracy of the target task, and the second term is the loss incurred by the adversary. It is worth pointing out that the minimax problem in (1) is *separable* for any fixed feature transformation $g$, in the sense that once $g$ has been fixed, the optimization of $f$ and $f'$ are independent of each other. Formally, define $R_Y^*(g) := \inf_f \mathbb{E}_{\mathcal{D}}\ell(f(g(X)), Y)$ to be the optimal risk in predicting

$Y$ using $Z = g(X)$ under loss $\ell$, and similarly define $R_A^*(g)$. The separation structure of the problem leads to the following compact form:

$$\mathbf{OPT}(\lambda) := \min_g R_Y^*(g) - \lambda \cdot R_A^*(g). \tag{2}$$

The minimization here is taken over a family of (possibly randomized) transformations $g$. Intuitively, (2) characterizes the situation where for a given transformation $Z = g(X)$, both $f$ and $f'$ play their optimal responses. Hence this objective function characterizes a fundamental limit of what the *best* possible representation we can hope to achieve for a fixed value $\lambda$. In general, with $0 < \lambda < \infty$, there is an inherent tension between the minimization of $R_Y^*(g)$ and the maximization of $R_A^*(g)$, and a choice of the tradeoff hyperparameter $\lambda$ essentially corresponds to a realization of such tradeoff.

**Motivating Examples** We discuss several examples to which the above framework is applicable.

**Example 2.1** (Privacy-Preservation)**.** In privacy applications, the goal is to make it difficult to predict sensitive data, represented by the attribute $A$, while retaining information about $Y$ (Hamm, 2015; 2017; Coavoux et al., 2018; Xiao et al., 2019). A way to achieve this is to pass information through $Z$, the "privatized" or "sanitized" data.

**Example 2.2** (Algorithmic Fairness)**.** In fairness applications, we seek to make predictions about the response $Y$ without discriminating based on the information contained in the protected attributes $A$. For example, $A$ may represent a protected class of individuals defined by, e.g. race or gender. This definition of fairness is also known as *statistical parity* in the literature, and has received increasing attention recently from an information-theoretic perspective (McNamara et al., 2019; Zhao & Gordon, 2019; Dutta et al., 2019).

**Example 2.3** (Domain Adaptation)**.** In domain adaptation, our goal is to train a predictor using labeled data from the source domain that generalizes to the target domain. In this case, $A$ corresponds to the identity of domains, and the hope here is to learn a domain-invariant representation $Z$ that is informative about the target $Y$ (Ben-David et al., 2007; 2010; Ganin et al., 2016; Zhao et al., 2018).

**Example 2.4** (Group Invariance)**.** In many applications in computer vision, it is desirable to learn predictors that are invariant to the action of a group $G$ on the input space. Typical examples include rotation, translation, and scale. By considering random variables $A$ that take their values in $G$, one approach to this problem is to learn a representation $Z$ that "ignores" changes in $A$ (Quiroga et al., 2005; Gens & Domingos, 2014; Bouvrie et al., 2009; Mallat, 2012; Anselmi et al., 2016).

**Example 2.5** (Information bottleneck)**.** The *information bottleneck* (Tishby et al., 2000) is the problem of finding a representation $Z$ that minimizes the objective $I(Z;Y) - \lambda I(Z;X)$ in an unsupervised manner. This is closely related to, but not the same as the problem we study, owing to the invariant attribute $A$.

## 3  FEASIBLE REGION ON THE INFORMATION PLANE

We begin by defining the *feasible region* associated with the adversarial game (1) and discussing its relevance to the problem we study. Formally, we define the *information plane* to be the 2D coordinate plane with axes $-R_Y^*(g)$ and $-R_A^*(g)$, respectively. The *feasible region* then corresponds to the two-dimensional region defined by the pairs $(-R_Y^*(g), -R_A^*(g))$ over all possible representations $Z = g(X)$ on the information plane. More concretely, in the classification and regression settings, these pairs can be given a more intuitive interpretation in terms of mutual information and conditional variances, respectively. In particular, it is easy to show the following:

1. (Classification) Under cross-entropy loss, using standard information-theoretic identities the adversarial game (2) can be rewritten as

$$\min_{Z=g(X)} H(Y \mid Z) - \lambda \cdot H(A \mid Z) \quad \Longleftrightarrow \quad \max_{Z=g(X)} I(Y;Z) - \lambda \cdot I(A;Z). \tag{3}$$

2. (Regression) Under the least-squares loss, by the law of total variance, the adversarial game (2) can be rewritten as

$$\min_{Z=g(X)} \mathbb{E}[\mathrm{Var}(Y \mid Z)] - \lambda \cdot \mathbb{E}[\mathrm{Var}(A \mid Z)] \quad \Longleftrightarrow \quad \max_{Z=g(X)} \mathrm{Var}\,\mathbb{E}[Y \mid Z] - \lambda \cdot \mathrm{Var}\,\mathbb{E}[A \mid Z]. \tag{4}$$

These equivalences motivate the following definitions:

$$\text{(Classification):} \quad \mathcal{R}_{\mathrm{CE}} := \{(I(Y;Z), I(A;Z)) \in \mathbb{R}^2\},$$

$$\text{(Regression):} \quad \mathcal{R}_{\mathrm{LS}} := \{(\mathrm{Var}\,\mathbb{E}[Y \mid Z], \mathrm{Var}\,\mathbb{E}[A \mid Z]) \in \mathbb{R}^2\}.$$

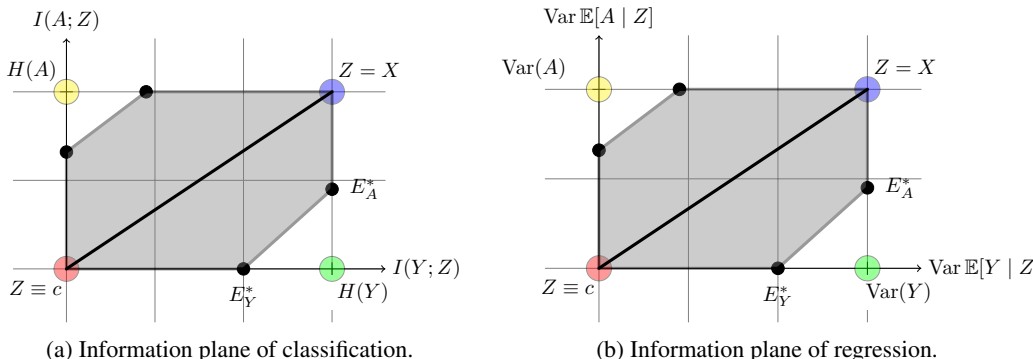

(a) Information plane of classification.  (b) Information plane of regression.

Figure 1: 2D information plane. The shaded area correspond to the feasible region.

We call both $\mathcal{R}_{\mathrm{CE}}$ and $\mathcal{R}_{\mathrm{LS}}$ the *feasible region* for the classification and regression settings, respectively. See Fig. 1 for an illustration of the information planes and the feasible regions. At this point, it may not be immediately clear what the relevance of the feasible region is. To see this, recall that our high-level goal is to find representations $Z$ that maximize accuracy (i.e. $\mathbb{E}_{\mathcal{D}}[\ell(f(Z), Y)]$) while simultaneously maximizing invariance (i.e. $\mathbb{E}_{\mathcal{D}}[\ell(f'(Z), A)]$), and consider the four vertices (not necessarily a part of the feasible region) in Fig. 1. These four corners have intuitive interpretations:

- (Red) The so-called "informationless" regime, in which all of the information regarding both $Y$ and $A$ is destroyed. This is achieved by choosing a constant representation $Z \equiv c$.
- (Yellow) Here, we retain all of the information in $A$ while removing all of the information about $Y$. This is not a particularly interesting regime for the aforementioned applications.
- (Blue) The full information regime, where $Z = X$ and no information is lost. This is the "classical" setting, wherein information about $A$ is allowed to leak into $Y$.
- (Green) This is the ideal representation that we would like to attain: we preserve all the relevant information about $Y$ while simultaneously removing all the information about $A$.

Unfortunately, in general, the ideal representation may not be attainable due to the potential correlation between $Y$ and $A$. As a result, we are interested in characterizing how "close" we can get to attaining this ideal transformation given the distribution over $(X, A, Y)$. More precisely, we can describe the various extremal points on the boundary of the feasible region as follows:

- $E_Y^*$: This point corresponds to a representation $Z$ that maximizes accuracy subject to a hard constraint on the invariance (cf. (5),(10)), i.e. there is no leakage of information about $A$ into the representation $Z$. In classification, we enforce this via the mutual information constraint $I(A; Z) = 0$ and in regression via the conditional variance constraint $\mathrm{Var}\,\mathbb{E}[A \mid Z] = 0$.
- $E_A^*$: This point corresponds to a representation $Z$ that maximizes invariance subject to a hard constraint on the accuracy (cf. (8), (12)), i.e. there is no loss of information about $Y$ in the representation $Z$. In classification, we enforce this via the mutual information constraint $I(Y; Z) = H(Y)$ and in regression we enforce this via the conditional variance constraint $\mathrm{Var}\,\mathbb{E}[Y \mid Z] = \mathrm{Var}(Y)$.
- As we vary $\lambda \in (0, \infty)$, we carve out a path $\mathbf{OPT}(\lambda)$ between $E_Y^*$ and $E_A^*$ that corresponds to the optimal values of (1). This is the Pareto frontier of the accuracy-invariance tradeoff, and represents the best possible tradeoff attainable for a given $\lambda$.

Due to the symmetry between $Y$ and $A$ in (2), the feasible regions in both cases are symmetric with respect to the diagonal of the bounding rectangle. With the feasible region more clearly exposed, we can now concretely outline our objective: To analytically characterize the solutions to the extremal problems corresponding to the lower and upper right points on the boundaries, and to provide lower bounds on the objective $\mathbf{OPT}(\lambda)$. Due to the page limit, we defer all the detailed proofs to appendix and mainly focus on providing interpretations and insights of our results in the main paper.

## 4  CLASSIFICATION

In order to understand the tradeoff between these two competing goals, it is the most interesting to study the case where the original input $X$ contains full information to predict both $Y$ and $A$, so that

any loss of accuracy is not due to the noninformative input $X$. To this end, our following analysis focuses on the *noiseless setting*[1]:

**Assumption 4.1.** There exist functions $f_Y^*(\cdot)$ and $f_A^*(\cdot)$, such that $Y = f_Y^*(X)$ and $A = f_A^*(X)$.

In order to characterize the feasible region $\mathcal{R}_{\mathrm{CE}}$, first note that from the data processing inequality, the following inequalities hold:

$$0 \leq I(Y; Z) \leq I(Y; X) = H(Y), \qquad 0 \leq I(A; Z) \leq I(A; X) = H(A),$$

which means that for any transformation $Z = g(X)$, the point $(I(Y; Z), I(A; Z))$ must lie within a rectangle shown in Fig. 2a. The following lemma shows that the feasible region $\mathcal{R}_{\mathrm{CE}}$ is convex:

**Lemma 4.1.** $\mathcal{R}_{\mathrm{CE}}$ is convex.

Here, the convexity of $\mathcal{R}_{\mathrm{CE}}$ is guaranteed by a construction of randomized feature transformation. As we briefly discussed before, we know that two vertices of the bounding rectangle are attainable, i.e., the "informationless" origin and the "full information" diagonal vertex. Now with Lemma 4.1, it is clear that all the points on the diagonal of the bounding rectangle are also attainable.

## 4.1 MAXIMAL MUTUAL INFORMATION UNDER THE INDEPENDENCE CONSTRAINT

In this section we explore the extremal point $E_Y^*$. This means that we would like to maximize the mutual information of $Z$ w.r.t $Y$ and simultaneously being independent of $A$:

$$\max_Z \quad I(Y; Z), \qquad \text{subject to} \quad I(A; Z) = 0. \tag{5}$$

First of all, realize that the optimal solution of (5) clearly depends on the coupling between $A$ and $Y$. To see this, consider the following two extreme cases:

**Example 4.1.** If $A = Y$ almost surely, then $I(A; Z) = 0$ directly implies $I(Y; Z) = 0$, hence $\max_Z I(Y; Z) = 0$ under the constraint that $I(A; Z) = 0$.

**Example 4.2.** If $A \perp Y$, then $Z = f_Y^*(X) = Y$ satisfies the constraint that $I(A; Z) = I(A; Y) = 0$. Furthermore, $I(Y; Z) = I(Y; Y) = H(Y) \geq I(Y; Z'), \forall Z' \neq Y$. Hence $\max_Z I(Y; Z) = H(Y)$.

The above two examples show that the optimal solution of (5), if exists analytically, must include a quantity that characterizes the dependency between $A$ and $Y$. We first define such a quantity:

**Definition 4.1.** Define $\Delta_{Y|A} := |\Pr_{\mathcal{D}}(Y = 1 \mid A = 0) - \Pr_{\mathcal{D}}(Y = 1 \mid A = 1)|$.

It is easy to verify that the following claims hold about $\Delta_{Y|A}$:

$$0 \leq \Delta_{Y|A} \leq 1, \text{ and } \Delta_{Y|A} = 0 \iff A \perp Y, \text{ and } \Delta_{Y|A} = 1 \iff A = Y \text{ or } A = 1 - Y. \tag{6}$$

With this introduced notation, the following theorem gives an analytic solution to (5).

**Theorem 4.1.** The optimal solution of optimization problem (5) is

$$\max_{Z, I(A; Z) = 0} I(Y; Z) = H(Y) - \Delta_{Y|A} \cdot H(A). \tag{7}$$

Let us have a sanity check of this result: First, if $A \perp Y$, then $\Delta_{Y|A} = 0$, and in this case the optimal solution given by Theorem 4.1 reduces to $H(Y) - 0 \cdot H(A) = H(Y)$, which is consistent with Example 4.2. Next, consider the other extreme case where $A = Y$. In this case $\Delta_{Y|A} = 1$ and $H(Y) = H(A)$, therefore the optimal solution given by Theorem 4.1 becomes $H(Y) - 1 \cdot H(A) = H(Y) - H(Y) = 0$. This is consistent with Example 4.1.

Moreover, due to the symmetry between $A$ and $Y$, we can now characterize the locations of the two extremal points on the lower and left boundaries. The updated figure is plotted in Fig. 2b. In Fig. 2b $\Delta_{A|Y}$ is defined analogously as $\Delta_{Y|A}$ by swapping $Y$ and $A$.

## 4.2 MINIMUM MUTUAL INFORMATION UNDER THE SUFFICIENT STATISTICS CONSTRAINT

Next, we characterize the other extremal point, i.e. $E_A^*$. Again, by the symmetry between $A$ and $Y$, it suffices to solve the following optimization problem, whose optimal solution is $E_A^*$.

$$\min_Z \quad I(A; Z), \qquad \text{subject to} \quad I(Y; Z) = H(Y) \tag{8}$$

---

[1]Extensions to the general noisy setting are feasible, but the results are less interpretable. Hence we mainly focus on the noiseless setting in this paper.

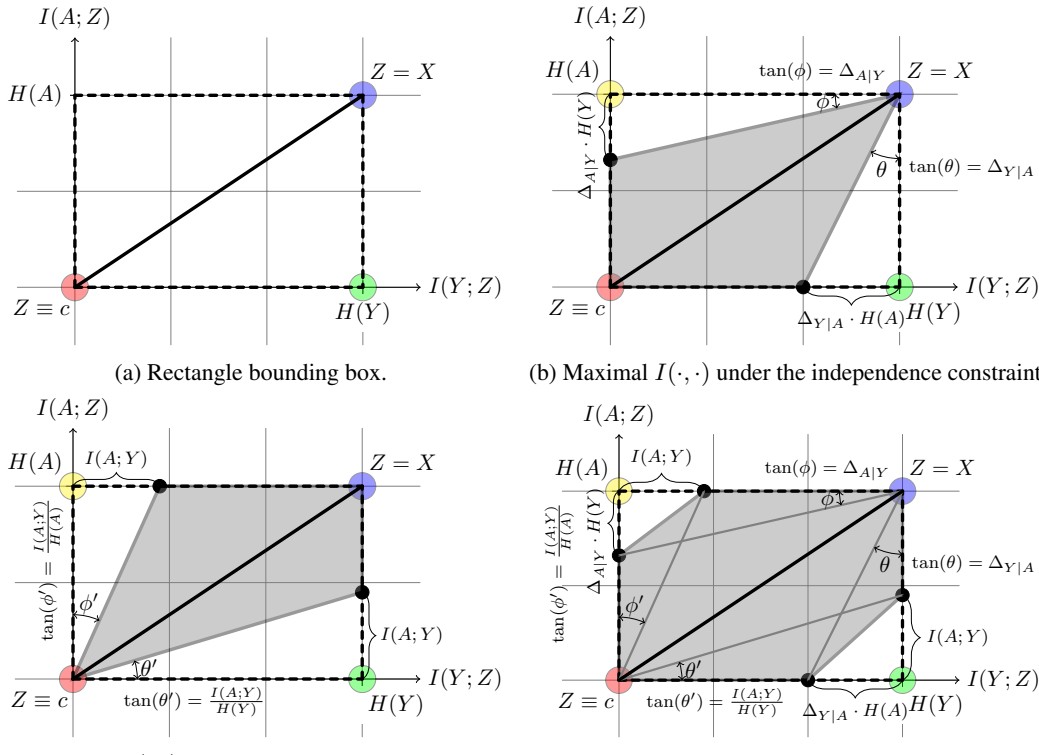

(a) Rectangle bounding box.

(b) Maximal $I(\cdot, \cdot)$ under the independence constraint.

(c) Minimum $I(\cdot, \cdot)$ under the sufficient-stat constraint.

(d) The convex polygon characterization of $\mathcal{R}_{\mathrm{CE}}$.

Figure 2: Information plane in classification. Shaded area corresponds to the known feasible region.

**Theorem 4.2.** The optimal solution of optimization problem (8) is

$$\min_{Z, I(Y;Z)=H(Y)} I(A; Z) = I(A; Y). \tag{9}$$

Clearly, if $A$ and $Y$ are independent, then the gap $I(A; Y) = 0$, meaning that we can simultaneously preserve all the target related information and filter out all the information related to $A$. With the above result, we can now characterize the locations of the remaining two extremal points on the top and right boundaries of bounding rectangle. The updated figure is shown in Fig. 2c.

### 4.3 THE INFORMATION PLANE IN LEARNING REPRESENTATIONS $Z$

To get the full picture, we combine our results in Section 4.1 and Section 4.2 and use the fact that $\mathcal{R}_{\mathrm{CE}}$ must be convex (Lemma 4.1). This allows us to complete the analysis by connecting the black dots on the four boundaries of the bounding rectangle, as shown in Fig. 2d. The feasible region $\mathcal{R}_{\mathrm{CE}}$ is a convex polygon. Furthermore, both the constrained accuracy optimal solution and the constrained invariance optimal solution can be readily read from Fig. 2d as well.

As we mentioned before, ideally we would like to find a representation $Z$ that attains the green vertex of the bounding rectangle. Unfortunately, due to the potential coupling between $Y$ and $A$, this solution is not always feasible. Nevertheless, it is instructive to see the gaps between the optimal solutions we could hope to achieve and the ideal one:

- Maximal information: The gap is given by $\Delta_{Y|A} \cdot H(A)$. On one hand, if $A \perp Y$, then $\Delta_{Y|A} = 0$ so the gap is 0. On the other hand, if $A = Y$, then $\Delta_{Y|A} = 1$ and $H(A) = H(Y)$, so the gap achieves the maximum value $H(Y)$.
- Maximal invariance: The gap is given by $I(A; Y)$. On one hand, if $A \perp Y$, then $I(A; Y) = 0$ so the gap is 0. On the other hand, if $A = Y$, then $I(A; Y) = H(A)$, so again, the gap achieves the maximum value of $H(A)$.

One open question that we do not answer here is whether the feasible region $\mathcal{R}_{\mathrm{CE}}$ is strictly convex or not. That is, whether the Pareto-frontier between $E_Y^*$ and $E_A^*$ is strictly convex or not? On the other hand, for each value of $\lambda$, the line segment connecting $E_Y^*$ and $E_A^*$ forms a lower bound of

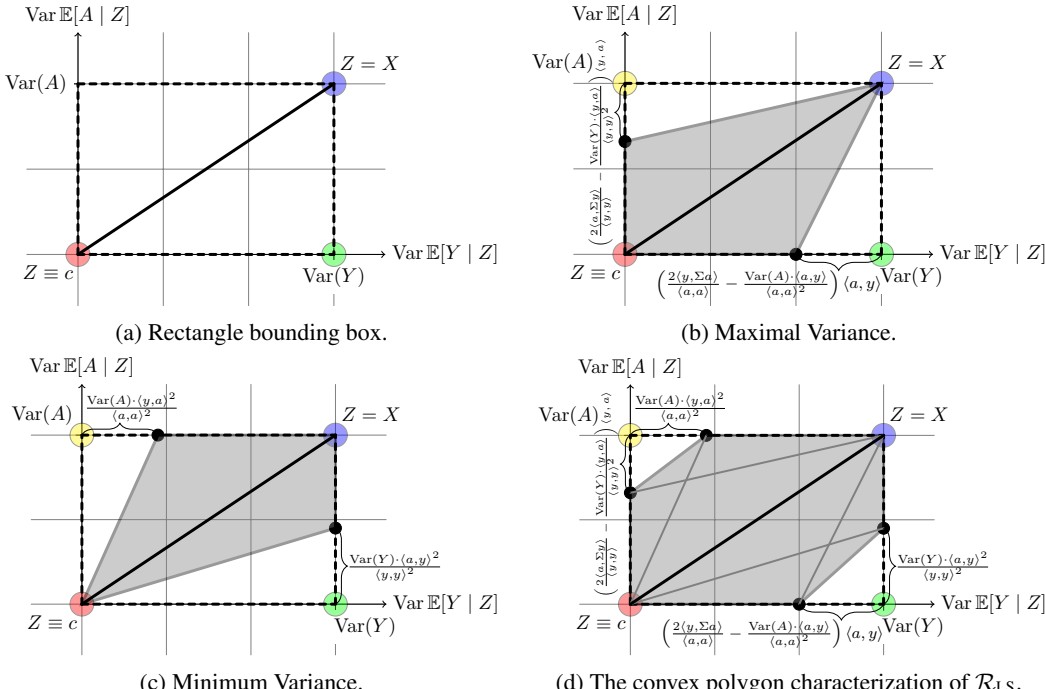

Figure 3: Information plane in regression. Shaded area corresponds to the known feasible region.

the Lagrangian. For the aforementioned applications, our approximation of the frontier is critical in order to be able to *certify* that a given model is *not* optimal. For example, given some practically computed representation $Z$ and by using known optimal estimators of the mutual information, it is possible to estimate $I(A; Y)$, $I(Y; Z)$, $I(A; Z)$ in order to directly bound the (sub)-optimality of $Z$ using Fig. 2d, i.e., how far away the point $(I(Y; Z), I(A; Z))$ is from the line segment between $E_Y^*$ and $E_A^*$. This distance lower bounds the distance to the optimal representations on the Pareto frontier.

## 5 REGRESSION

Similar to what we have before, in regression we assume a noiseless setting for better interpretability of our results. The generalization to the noisy setting is included in the appendix. Let $\mathbb{H}$ be an RKHS.

**Assumption 5.1.** There exist functions $f_Y^*, f_A^* \in \mathbb{H}$, such that $Y = f_Y^*(X)$ and $A = f_A^*(X)$.

Let $\langle \cdot, \cdot \rangle$ be the canonical inner product in RKHS $\mathbb{H}$. Under this assumption, there exists a feature map $\varphi(X)$ and $a \neq 0, y \neq 0$, such that $Y = f_Y^*(X) = \langle \varphi(X), y \rangle$ and $A = f_A^*(X) = \langle \varphi(X), a \rangle$. This feature map does not have to be finite-dimensional, and our analysis works for the case where $f_Y^*, f_A^*$ are infinite-dimensional. Next, by the law of total variance, the following inequalities hold:

$$0 \leq \operatorname{Var} \mathbb{E}[Y \mid Z] \leq \operatorname{Var} \mathbb{E}[Y \mid X] = \operatorname{Var}(Y), \qquad 0 \leq \operatorname{Var} \mathbb{E}[A \mid Z] \leq \operatorname{Var} \mathbb{E}[A \mid X] = \operatorname{Var}(A),$$

which means that for any transformation $Z = g(X)$, the point $(\operatorname{Var} \mathbb{E}[Y \mid Z], \operatorname{Var} \mathbb{E}[A \mid Z])$ must lie within a rectangle shown in Fig. 3a. To simplify the notation, we define $\Sigma := \operatorname{Cov}(\varphi(X), \varphi(X))$ to be the covariance operator of $\varphi(X)$.

Again, if we consider all the possible feature transformations $Z = g(X)$, then the points $(\operatorname{Var} \mathbb{E}[Y \mid Z], \operatorname{Var} \mathbb{E}[A \mid Z])$ will form a feasible region $\mathcal{R}_{\mathrm{LS}}$. Similar to what we have in classification, the following lemma shows that the feasible region $\mathcal{R}_{\mathrm{LS}}$ is convex:

**Lemma 5.1.** $\mathcal{R}_{\mathrm{LS}}$ is convex.

The convexity of $\mathcal{R}_{\mathrm{LS}}$ is guaranteed by a construction of randomized feature transformation. Similarly, both the "informationless" origin and the "full information" diagonal vertex are attainable.

### 5.1 THE BOUNDING VERTICES ON THE PLANE

In this section we explore the extremal point $E_Y^*$ and $E_A^*$ for regression. For $E_Y^*$, this means that we would like to maximize the variance of $Z$ w.r.t $Y$ and simultaneously minimizing that of $A$:

$$\max_Z \quad \operatorname{Var}\mathbb{E}[Y \mid Z], \qquad \text{subject to} \quad \operatorname{Var}\mathbb{E}[A \mid Z] = 0. \tag{10}$$

It is clear that the optimal solution of (10) depends on the coupling between $A$ and $Y$, and the following theorem precisely characterizes this relationship:

**Theorem 5.1.** The optimal solution of optimization problem (10) is upper bounded by

$$\max_{Z, \operatorname{Var}\mathbb{E}[A|Z]=0} \operatorname{Var}\mathbb{E}[Y \mid Z] \leq \operatorname{Var}(Y) - \left( \frac{2\langle y, \Sigma a \rangle}{\langle a, a \rangle} - \frac{\operatorname{Var}(A) \cdot \langle a, y \rangle}{\langle a, a \rangle^2} \right) \langle a, y \rangle. \tag{11}$$

Again, let us sanity check this result: First, if $a$ is orthogonal to $y$, i.e., $\langle a, y \rangle = 0$, then the gap is 0, and the optimal solution becomes $\operatorname{Var}(Y)$. Next, consider the other extreme case where $a$ is parallel to $y$. In this case it can be readily verified that the optimal solution reduces to 0. With these two results, we can now characterize the locations of the two extremal points on the bottom and left boundaries of bounding rectangle. The updated figure is plotted in Fig. 3b.

Similarly, for $E_A^*$, it suffices to solve the following problem, whose optimal solution is $E_A^*$:

$$\min_Z \quad \operatorname{Var}\mathbb{E}[A \mid Z], \qquad \text{subject to} \quad \operatorname{Var}\mathbb{E}[Y \mid Z] = \operatorname{Var}(Y) \tag{12}$$

**Theorem 5.2.** The optimal solution of optimization problem (12) is lower bounded by

$$\min_{Z, \operatorname{Var}\mathbb{E}[Y|Z] \geq \operatorname{Var}(Y)} \operatorname{Var}\mathbb{E}[A \mid Z] \geq \frac{\operatorname{Var}(Y) \cdot \langle a, y \rangle^2}{\langle y, y \rangle^2}. \tag{13}$$

Again, if $a$ is orthogonal to $y$, then the optimal solution is 0, meaning that we can simultaneously preserve all the target variance and filter out all the variance related to $A$. On the other hand, if $a$ is parallel to $y$, then $\operatorname{Var}(Y) \cdot \langle a, y \rangle^2 / \langle y, y \rangle^2 = \operatorname{Var}(A)$. The updated plot is shown in Fig. 3c.

### 5.2 A SPECTRAL LOWER BOUND OF THE LAGRANGIAN

Combining our results in Thm. 5.1 and Thm. 5.2, along with the fact that $\mathcal{R}_{\mathrm{LS}}$ must be convex, we plot the full picture about the feasible region in the regression setting in Fig. 3d. Both the constrained accuracy optimal solution and the constrained invariance optimal solution can be readily read from Fig. 3d as well. In fact, in the regression setting, we can say even more: We can derive a tight lower bound to the Lagrangian problem $\mathbf{OPT}(\lambda) := \min_{Z=g(X)} \mathbb{E}[\operatorname{Var}(Y \mid Z)] - \lambda \cdot \mathbb{E}[\operatorname{Var}(A \mid Z)]$.

**Theorem 5.3.** The optimal solution of the Lagrangian has the following lower bound:

$$\mathbf{OPT}(\lambda) \geq \frac{1}{2} \left\{ \operatorname{Var}(Y) - \lambda \cdot \operatorname{Var}(A) - \sqrt{(\operatorname{Var}(Y) + \lambda \cdot \operatorname{Var}(A))^2 - 4\lambda \langle y, \Sigma a \rangle^2} \right\}. \tag{14}$$

Evidently, the key quantity in the lower bound (14) is the quadratic term $\langle y, \Sigma a \rangle$, which effectively measures the dependence between the $Y$ and $A$ under the feature covariance $\Sigma$.

The proof of Thm. 5.1, 5.2 and 5.3 rely on a finite-dimensional SDP relaxation, and constructs an explicit optimal solution to this relaxation. We re-formulate the objective as a linear functional of $V := \operatorname{Cov}(\mathbb{E}[\varphi(X) \mid Z], \mathbb{E}[\varphi(X) \mid Z])$, which satisfies the semi-definite constraint $0 \preceq V \preceq \Sigma = \operatorname{Cov}(\varphi(X), \varphi(X))$. Therefore, the optimal value of the SDP is an upper/lower bound of the objective. Furthermore, we show that under certain regularity conditions, the SDP relaxation is *exact*. One particularly interesting setting where the regularity condition holds is when $\varphi(X)$ follows a Gaussian distribution. More discussions about the tightness of our bounds are presented in appendix C.5.

## 6 CONCLUSION

We provide an information plane analysis to study the general and important problem for learning invariant representations in both classification and regression settings. In both cases, we analyze the inherent tradeoffs between accuracy and invariance by providing a geometric characterization of the feasible region on the information plane, in terms of its boundedness, convexity, as well as its its extremal vertices. Furthermore, in the regression setting, we also derive a tight lower bound that for the Lagrangian form of accuracy and invariance. Given the wide applications of invariant representations in machine learning, we believe our theoretical results could contribute to better understandings of the fundamental tradeoffs between accuracy and invariance under various settings, e.g., domain adaptation, algorithmic fairness, invariant visual representations, and privacy-preservation learning.

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

# A    PROOFS FOR CLAIMS IN SECTION 3

In this section we give detailed arguments to derive the objective functions of Eq. (3) and (4) respectively from the original minimax formulation in Eq. (1). First, let us consider the classification setting.

**Classification**    Given a fixed feature map $Z = g(X)$, due to the symmetry between $Y$ and $A$ in Eq. (1), it suffices for us to consider the case of finding $f$ that minimizes $\mathbb{E}_{\mathcal{D}}[\ell(f \circ g(X), Y)]$, and analogous result follows for the case of finding the optimal $f'$ that minimizes $\mathbb{E}_{\mathcal{D}}[\ell(f' \circ g(X), A)]$ similarly. By definition of the cross-entropy loss, we have:

$$
\begin{aligned}
\mathbb{E}_{\mathcal{D}}[\ell(f \circ g(X), Y)] &= -\mathbb{E}_{\mathcal{D}}\left[\mathbb{I}(Y = 0)\log(1 - f(g(X))) + \mathbb{I}(Y = 1)\log(g(f(X)))\right] \\
&= -\mathbb{E}_{\mathcal{D}}\left[\mathbb{I}(Y = 0)\log(1 - f(Z)) + \mathbb{I}(Y = 1)\log(f(Z))\right] \\
&= -\mathbb{E}_{Z}\mathbb{E}_{Y}\left[\mathbb{I}(Y = 0)\log(1 - f(Z)) + \mathbb{I}(Y = 1)\log(f(Z)) \mid Z\right] \\
&= -\mathbb{E}_{Z}\left[\Pr(Y = 0 \mid Z)\log(1 - f(Z)) + \Pr(Y = 1 \mid Z)\log(f(Z))\right] \\
&= \mathbb{E}_{Z}\left[D_{\mathrm{KL}}(\Pr(Y \mid Z) \parallel f(Z))\right] + H(Y \mid Z) \\
&\geq H(Y \mid Z).
\end{aligned}
$$

It is also clear from the above proof that the minimum value of the cross-entropy loss is achieved when $f(Z)$ is a randomized classifier such that $\mathbb{E}[f(Z)] = \Pr(Y = 1 \mid Z)$. This shows that

$$
\min_{f} \mathbb{E}_{\mathcal{D}}[\ell(f \circ g(X), Y)] = H(Y \mid Z), \quad \text{and} \quad \min_{f'} \mathbb{E}_{\mathcal{D}}[\ell(f' \circ g(X), A)] = H(A \mid Z).
$$

To see the second part of Eq. (3), simply use the identity that $H(Y \mid Z) = H(Y) - I(Y; Z)$ and $H(A \mid Z) = H(A) - I(A; Z)$ with the fact that both $H(Y)$ and $H(A)$ are constants that only depend on the joint distribution $\mathcal{D}$.

**Regression**    Again, given a fixed feature map $Z = g(X)$, because of the symmetry between $Y$ and $A$ let us focus on the analysis of finding $f$ that minimizes $\mathbb{E}_{\mathcal{D}}[\ell(f \circ g(X), Y)]$. In this case since $\ell(\cdot, \cdot)$ is the mean squared error, it follows that

$$
\begin{aligned}
\mathbb{E}_{\mathcal{D}}[\ell(f \circ g(X), Y)] &= \mathbb{E}_{\mathcal{D}}\left[(f \circ g(X) - Y)^2\right] \\
&= \mathbb{E}_{\mathcal{D}}\left[(f(Z) - Y)^2\right] \\
&= \mathbb{E}_{Z}\left[(f(Z) - \mathbb{E}[Y \mid Z])^2\right] + \mathbb{E}_{Z}\left[\mathbb{E}_{Y}[(Y - \mathbb{E}[Y \mid Z])^2]\right] \\
&\geq \mathbb{E}_{Z}\left[\mathbb{E}_{Y}[(Y - \mathbb{E}[Y \mid Z])^2]\right] \\
&= \mathbb{E}[\mathrm{Var}(Y \mid Z)],
\end{aligned}
$$

where the third equality is due to the Pythagorean theorem. Furthermore, it is clear that the optimal mean-squared error is obtained by the conditional mean $f(Z) = \mathbb{E}[Y \mid Z]$. This shows that

$$
\min_{f} \mathbb{E}_{\mathcal{D}}[\ell(f \circ g(X), Y)] = \mathbb{E}[\mathrm{Var}(Y \mid Z)], \quad \text{and} \quad \min_{f'} \mathbb{E}_{\mathcal{D}}[\ell(f' \circ g(X), A)] = \mathbb{E}[\mathrm{Var}(A \mid Z)].
$$

For the second part, use the law of total variance $\mathrm{Var}(Y) = \mathbb{E}[\mathrm{Var}(Y \mid Z)] + \mathrm{Var}(\mathbb{E}[Y \mid Z])$ and $\mathrm{Var}(A) = \mathbb{E}[\mathrm{Var}(A \mid Z)] + \mathrm{Var}(\mathbb{E}[A \mid Z])$. Realizing that both $\mathrm{Var}(Y)$ and $\mathrm{Var}(A)$ are constants that only depend on the joint distribution $\mathcal{D}$, we finish the proof.

# B    MISSING PROOFS IN CLASSIFICATION (SECTION 4)

In what follows we first restate the propositions, lemmas and theorems in the main text, and then provide the corresponding proofs.

## B.1    CONVEXITY OF $\mathcal{R}_{\mathrm{CE}}$

**Lemma 4.1.** $\mathcal{R}_{\mathrm{CE}}$ is convex.

*Proof.* Let $Z_i = g_i(X)$ for $i \in \{0, 1\}$ with corresponding points $(I(Y; Z_i), I(A; Z_i)) \in \mathcal{R}_{\mathrm{CE}}$. Then we only need to prove that for $\forall u \in [0, 1]$, $(uI(Y; Z_0) + (1 - u)I(Y; Z_1), uI(A; Z_0) + (1 -$

$u)I(A; Z_1)) \in \mathcal{R}_{CE}$ as well. For any $u \in [0, 1]$, let $S \sim U(0, 1)$, the uniform distribution over $(0, 1)$, such that $S \perp (Y, A)$. Consider the following randomized transformation $Z$:

$$Z = \begin{cases} Z_0 & \text{If } S \leq u, \\ Z_1 & \text{otherwise.} \end{cases} \tag{15}$$

To compute $I(Y; Z)$, we have:

$$\begin{aligned} I(Y; Z) &= \mathbb{E}[I(Y; Z \mid S)] \\ &= \Pr(S \leq u) \cdot I(Y; Z_0) + \Pr(S > u) \cdot I(Y; Z_1) \\ &= u I(Y; Z_0) + (1 - u) I(Y; Z_1). \end{aligned}$$

Similar argument could be used to show that $I(A; Z) = u I(A; Z_0) + (1 - u) I(A; Z_1)$. So by construction we now find a randomized transformation $Z = g(X)$ such that $(u I(Y; Z_0) + (1 - u) I(Y; Z_1), u I(A; Z_0) + (1 - u) I(A; Z_1)) \in \mathcal{R}_{CE}$. ∎

### B.2 PROOF OF THEOREM 4.1

We proceed to provide the proof that the optimal value of (5) is the one given by Theorem 4.1.

**Theorem 4.1.** The optimal solution of optimization problem (5) is

$$\max_{Z, I(A;Z)=0} I(Y; Z) = H(Y) - \Delta_{Y|A} \cdot H(A). \tag{7}$$

*Proof.* For a joint distribution $\mathcal{D}$ over $(X, A, Y)$ and a function $g : \mathcal{X} \to \mathcal{Z}$, in what follows we use $g_\sharp \mathcal{D}$ to denote the induced distribution of $\mathcal{D}$ under $g$ over $(Z, A, Y)$. We first make the following claim: without loss of generality, for any joint distribution $g_\sharp \mathcal{D}$ over $(Z, A, Y)$, we could find $(Z_0, A', Y') \sim g_\sharp \mathcal{D}$ and a deterministic function $f$, such that $Y' = f(A', Z_0, S)$ where $S \sim U(0, 1)$, $S \perp (A', Z_0)$ and $I(Y'; Z') \geq I(Y; Z)$ with $Z' = (Z_0, S)$. To see this, consider the following construction:

$$A', Z_0 \sim \mathcal{D}(A, Z), \quad S \sim U(0, 1).$$

Let $(a, z, s)$ be the sample of the above sampling process and construct

$$Y' = \begin{cases} 1 & \text{If } s \leq \mathbb{E}[Y \mid A = a, Z = z], \\ 0 & \text{Otherwise.} \end{cases}$$

Now it is easy to verify that $(Z_0, A', Y') \sim g_\sharp \mathcal{D}$ and $\Pr(Y' = 1 \mid A' = a, Z_0 = z) = \mathbb{E}[Y \mid A = a, Z = z]$. To see the last claim, we have the following inequality hold:

$$I(Y'; Z') = I(Y'; Z_0, S) \geq I(Y; Z_0) = I(Y; Z).$$

Now to upper bound $I(Y; Z)$, we have

$$I(Y; Z) = H(Y) - H(Y \mid Z),$$

hence it suffices to lower bound $H(Y \mid Z)$. To this end, define

$$\begin{aligned} D_0 &:= \{z, \varepsilon \in (0, 1) \mid f(0, z, \varepsilon) = 1\}, \\ D_1 &:= \{z, \varepsilon \in (0, 1) \mid f(1, z, \varepsilon) = 1\}. \end{aligned}$$

Then,

$$\begin{aligned} \Pr((z, \varepsilon) \in D_0) &= \Pr(f(0, z, \varepsilon) = 1) \\ &= \Pr(f(0, z, \varepsilon) = 1 \mid A = 0) \\ &= \Pr(f(A, z, \varepsilon) = 1 \mid A = 0) \\ &= \Pr(Y = 1 \mid A = 0). \end{aligned}$$

Analogously, the following equation also holds:

$$\Pr((z, \varepsilon) \in D_1) = \Pr(Y = 1 \mid A = 1).$$

Without loss of generality, assume that $\Pr(Y = 1 \mid A = 1) \geq \Pr(Y = 1 \mid A = 0)$, then
$$\Pr((z, \varepsilon) \in D_1 \backslash D_0) \geq \Pr((z, \varepsilon) \in D_1) - \Pr((z, \varepsilon) \in D_0)$$
$$= |\Pr(Y = 1 \mid A = 1) - \Pr(Y = 1 \mid A = 0)|.$$
But on the other hand, we know that if $(z, \varepsilon) \in D_1 \backslash D_0$, then $f(1, z, \varepsilon) = 1$ and $f(0, z, \varepsilon) = 0$, and this implies that $Y = A$, hence:
$$H(Y \mid Z) \geq H(Y \mid Z, S)$$
$$= \Pr((z, \varepsilon) \in D_1 \backslash D_0) \cdot H(Y \mid (z, \varepsilon) \in D_1 \backslash D_0) + \Pr((z, \varepsilon) \notin D_1 \backslash D_0) \cdot H(Y \mid (z, \varepsilon) \notin D_1 \backslash D_0)$$
$$\geq \Pr((z, \varepsilon) \in D_1 \backslash D_0) \cdot H(Y \mid (z, \varepsilon) \in D_1 \backslash D_0)$$
$$= \Pr((z, \varepsilon) \in D_1 \backslash D_0) \cdot H(A)$$
$$\geq |\Pr(Y = 1 \mid A = 1) - \Pr(Y = 1 \mid A = 0)| \cdot H(A),$$
which implies that
$$I(Y; Z) \leq H(Y) - |\Pr(Y = 1 \mid A = 1) - \Pr(Y = 1 \mid A = 0)| \cdot H(A) = H(Y) - \Delta_{Y|A} \cdot H(A).$$
To see that the upper bound could be attained, let us consider the following construction. Denote $\alpha := \Pr(Y = 1 \mid A = 0)$ and $\beta := \Pr(Y = 1 \mid A = 1)$. Construct a uniformly random $Z \sim U(0, 1)$ and then sample $A$ independently from $Z$ according to the corresponding marginal distribution of $A$ in $\mathcal{D}$. Next, define:
$$Y = \begin{cases} 1 & \text{if } Z \leq \alpha \wedge A = 0 \text{ or } Z \leq \beta \wedge A = 1, \\ 0 & \text{otherwise.} \end{cases}$$
It is easy to see that $Z \perp A$ by construction. Furthermore, by the construction of $Y$, we also have $A, Y \sim \mathcal{D}(A, Y)$ hold. Since $I(Y; Z) = H(Y) - H(Y \mid Z)$, we only need to verify $H(Y \mid Z) = \Delta_{Y|A} \cdot H(A)$ in this case. Assume without loss of generality $\alpha \leq \beta$, there are three different cases depending on the value of $Z$:

- $Z \leq \alpha$: In this case no matter what the value of $A$, we always have $Y = 1$.
- $Z > \beta$: In this case no matter what the value of $A$, we always have $Y = 0$.
- $\alpha < Z \leq \beta$: In this case $Y = A$, hence the conditional distribution of $Y$ given $Z \in (\alpha, \beta]$ is equal to the conditional distribution of $A$ given $Z \in (\alpha, \beta]$. But by our construction, $A$ is independent of $Z$, which means that in this case the conditional distribution of $A$ given $Z \in (\alpha, \beta]$ is just the distribution of $A$.

Combine all the above three cases, we have:
$$H(Y \mid Z) = \Pr(Z \leq \alpha) \cdot H(Y \mid Z \leq \alpha) + \Pr(Z > \beta) \cdot H(Y \mid Z > \beta) + \Pr(\alpha < Z \leq \beta) \cdot H(Y \mid \alpha < Z \leq \beta)$$
$$= 0 + 0 + |\beta - \alpha| \cdot H(A \mid \alpha < Z \leq \beta)$$
$$= |\Pr(Y = 1 \mid A = 1) - \Pr(Y = 1 \mid A = 0)| \cdot H(A)$$
$$= \Delta_{Y|A} \cdot H(A),$$
which completes the proof. ∎

### B.3 Proof of Theorem 4.2

**Theorem 4.2.** The optimal solution of optimization problem (8) is
$$\min_{Z, I(Y;Z) = H(Y)} I(A; Z) = I(A; Y). \tag{9}$$

*Proof.* First, realize that $H(Z) \geq I(Y; Z) = H(Y)$ by our constraint. Furthermore, we also know that $0 \leq H(Y \mid Z, A) \leq H(Y \mid Z) = H(Y) - I(Y; Z) = 0$, which means $H(Y \mid Z, A) = 0$. With these two observations, we have:
$$I(A; Z) = H(Z) - H(Z \mid A)$$
$$\geq H(Y) - H(Z \mid A)$$
$$\geq H(Y) - H(Y, Z \mid A)$$
$$= H(Y) - H(Y \mid A) - H(Y \mid Z, A)$$
$$= H(Y) - H(Y \mid A)$$
$$= I(A; Y).$$
To attain the equality, simply set $Z = f_Y^*(X) = Y$. Specifically, this implies that 1-bit is sufficient to encode all the information for the optimal solution, which completes the proof. ∎

## C  MISSING PROOFS IN REGRESSION (SECTION 5)

### C.1  CONVEXITY OF $\mathcal{R}_{\mathrm{LS}}$

Analogous to the classification setting, here we first show that the feasible region $\mathcal{R}_{\mathrm{LS}}$ is convex:

**Lemma 5.1.** $\mathcal{R}_{\mathrm{LS}}$ is convex.

*Proof.* Let $Z_i = g_i(X)$ for $i \in \{0, 1\}$ with corresponding points $(\mathrm{Var}\,\mathbb{E}[Y \mid Z_i], \mathrm{Var}\,\mathbb{E}[A \mid Z_i]) \in \mathcal{R}_{\mathrm{LS}}$. Then it suffices if we could show for $\forall u \in [0, 1]$, $(u\,\mathrm{Var}\,\mathbb{E}[Y \mid Z_0] + (1 - u)\,\mathrm{Var}\,\mathbb{E}[Y \mid Z_1], u\,\mathrm{Var}\,\mathbb{E}[A \mid Z_0]) + (1 - u)\,\mathrm{Var}\,\mathbb{E}[A \mid Z_1]) \in \mathcal{R}_{\mathrm{LS}}$ as well.

We give a constructive proof. Due to the symmetry between $A$ and $Y$, we will only prove the result for $Y$, and the same analysis could be directly applied to $A$ as well. For any $u \in [0, 1]$, let $U \sim U(0, 1)$, the uniform distribution over $(0, 1)$, such that $U \perp (Y, A)$. Consider the following randomized transformation $Z$:

$$Z = \begin{cases} Z_0 & \text{If } U \leq u, \\ Z_1 & \text{otherwise.} \end{cases} \tag{16}$$

To compute $\mathrm{Var}\,\mathbb{E}[Y \mid Z]$, define $K := \mathbb{E}[Y \mid Z]$, then by the law of total variance, we have:

$$\mathrm{Var}\,\mathbb{E}[Y \mid Z] = \mathrm{Var}(K) = \mathbb{E}[\mathrm{Var}(K \mid U)] + \mathrm{Var}\,\mathbb{E}[K \mid U].$$

We first compute $\mathrm{Var}\,\mathbb{E}[K \mid U]$:

$$\begin{aligned}
\mathrm{Var}\,\mathbb{E}[K \mid U] &= \mathrm{Var}\,\mathbb{E}[\mathbb{E}[Y \mid Z] \mid U] \\
&= \mathrm{Var}\,\mathbb{E}[Y \mid U] &&\text{(The law of total expectation)} \\
&= \mathrm{Var}\,\mathbb{E}[Y] &&(Y \perp U) \\
&= 0.
\end{aligned}$$

On the other hand, for $\mathbb{E}[\mathrm{Var}(K \mid U)]$, we have:

$$\begin{aligned}
\mathbb{E}[\mathrm{Var}(K \mid U)] &= \Pr(U = 0) \cdot \mathrm{Var}(K \mid U = 0) + \Pr(U = 1) \cdot \mathrm{Var}(K \mid U = 1) \\
&= u \cdot \mathrm{Var}(K \mid U = 0) + (1 - u) \cdot \mathrm{Var}(K \mid U = 1) \\
&= u \cdot \mathrm{Var}\,\mathbb{E}[Y \mid Z_0] + (1 - u) \cdot \mathrm{Var}\,\mathbb{E}[Y \mid Z_1].
\end{aligned}$$

Combining both equations above yields:

$$\mathrm{Var}\,\mathbb{E}[Y \mid Z] = u \cdot \mathrm{Var}\,\mathbb{E}[Y \mid Z_0] + (1 - u) \cdot \mathrm{Var}\,\mathbb{E}[Y \mid Z_1].$$

Similar argument could be used to show that $\mathrm{Var}\,\mathbb{E}[A \mid Z] = u \cdot \mathrm{Var}\,\mathbb{E}[A \mid Z_0] + (1 - u) \cdot \mathrm{Var}\,\mathbb{E}[A \mid Z_1]$. So by construction we now find a randomized transformation $Z = g(X)$ such that $(u \cdot \mathrm{Var}\,\mathbb{E}[Y \mid Z_0] + (1 - u) \cdot \mathrm{Var}\,\mathbb{E}[Y \mid Z_1], u \cdot \mathrm{Var}\,\mathbb{E}[A \mid Z_0] + (1 - u) \cdot \mathrm{Var}\,\mathbb{E}[A \mid Z_1]) \in \mathcal{R}_{\mathrm{LS}}$, which completes the proof. ∎

### C.2  PROOF OF THEOREM 5.1 AND THEOREM 5.2

In this section, we will prove Theorem 5.1 and Theorem 5.2. We will provide proofs to both theorems in a generalized noisy setting, i.e., we no longer assume the noiseless condition so that the corresponding theorems in the noiseless setting follow as a special case. To this end, we first re-define

$$f_Y^*(X) := \mathbb{E}[Y \mid X] \tag{17}$$
$$f_A^*(X) := \mathbb{E}[A \mid X] \tag{18}$$

and $f_Y^*, f_A^* \in \mathbb{H}$. We reuse the notations $a, y$ to denote

$$f_Y^*(X) = \mathbb{E}[Y|X] = \langle y, \varphi(X) \rangle \tag{19}$$
$$f_A^*(X) = \mathbb{E}[A|X] = \langle a, \varphi(X) \rangle. \tag{20}$$

It is easy to see that the noiseless setting is indeed a special case where $Y = \mathbb{E}[Y|X], A = \mathbb{E}[A|X]$ almost surely.

For readers' convenience, we restate the Theorem 5.1 below:

**Theorem 5.1.** The optimal solution of optimization problem (10) is upper bounded by

$$\max_{Z, \text{Var} \, \mathbb{E}[A|Z]=0} \text{Var} \, \mathbb{E}[Y \mid Z] \leq \text{Var}(Y) - \left( \frac{2\langle y, \Sigma a \rangle}{\langle a, a \rangle} - \frac{\text{Var}(A) \cdot \langle a, y \rangle}{\langle a, a \rangle^2} \right) \langle a, y \rangle. \tag{11}$$

The following theorem is the generalized version of Theorem 5.1 in noisy setting:

**Theorem C.1.** The optimal solution of optimization problem (10) is upper bounded by

$$\max_{Z, \text{Var} \, \mathbb{E}[A|Z]=0} \text{Var} \, \mathbb{E}[Y \mid Z] \leq \text{Var}(\mathbb{E}[Y|X]) - \left( \frac{2\langle y, \Sigma a \rangle}{\langle a, a \rangle} - \frac{\text{Var}(\mathbb{E}[A|X]) \cdot \langle a, y \rangle}{\langle a, a \rangle^2} \right) \langle a, y \rangle. \tag{21}$$

It is easy to see Theorem 5.1 is an immediate corollary of this result: under the noiseless assumption, we have $\text{Var}(\mathbb{E}[Y|X]) = \text{Var}(Y)$ and $\text{Var}(\mathbb{E}[A|X]) = \text{Var}(A)$.

*Proof.* Using the law of total expectation,

$$\mathbb{E}[Y \mid Z] = \mathbb{E}\left[\mathbb{E}[Y \mid X] \mid Z\right] = \int_{\mathcal{X}} \mathbb{E}[Y \mid X, Z] \cdot p(X \mid Z) \, dX.$$

Since $Z = g(X)$ is a function of $X$, we have $Z \perp Y \mid X$, so $\mathbb{E}[Y \mid X, Z] = \mathbb{E}[Y \mid X] = f_Y^*(X)$. Therefore,

$$\begin{aligned} \mathbb{E}[Y|Z] &= \int_{\mathcal{X}} \mathbb{E}[Y \mid X, Z] \cdot p(X \mid Z) \, dX \\ &= \int_{\mathcal{X}} f_Y^*(X) \cdot p(X \mid Z) \, dX \\ &= \mathbb{E}[f_Y^*(X) \mid Z]. \end{aligned}$$

Hence,

$$\text{Var}(\mathbb{E}[Y \mid Z]) = \text{Var}(\mathbb{E}[f_Y^*(X) \mid Z]). \tag{22}$$

Therefore,

$$\begin{aligned} \text{Var} \, \mathbb{E}[Y \mid Z] &= \text{Var}(\mathbb{E}[f_Y^*(X) \mid Z]) \\ &= \text{Var} \, \mathbb{E}[\langle y, \varphi(X) \rangle \mid Z] \\ &= \text{Var}\langle y, \mathbb{E}[\varphi(X) \mid Z]\rangle \qquad \text{(Linearity of Expectation)} \\ &= \langle y, \text{Cov}(\mathbb{E}[\varphi(X) \mid Z], \mathbb{E}[\varphi(X) \mid Z])y\rangle. \end{aligned}$$

Similarly, for $A = \langle a, \varphi(X) \rangle$, we have:

$$\text{Var} \, \mathbb{E}[A \mid Z] = \langle a, \text{Cov}(\mathbb{E}[\varphi(X) \mid Z], \mathbb{E}[\varphi(X) \mid Z])a\rangle.$$

To simplify the notation, define $V := \text{Cov}(\mathbb{E}[\varphi(X) \mid Z], \mathbb{E}[\varphi(X) \mid Z])$. Then again, by the law of total variance, it is easy to verify that $0 \preceq V \preceq \Sigma = \text{Cov}(\varphi(X), \varphi(X))$. Hence the original maximization problem could be relaxed as follows:

$$\max_{Z} \quad \langle y, Vy \rangle, \qquad \text{subject to} \quad 0 \preceq V \preceq \Sigma, \langle a, Va \rangle = 0.$$

To proceed, we first decompose $y$ orthogonally w.r.t. $a$:

$$y = y^{\perp a} + y^{\| a},$$

where $y^{\perp a}$ is the component of $y$ that is perpendicular to $a$ and $y^{\| a}$ is the parallel component of $y$ to $a$. Using this orthogonal decomposition, we have $\forall V$:

$$\begin{aligned} \langle y, Vy \rangle &= \langle (y^{\perp a} + y^{\| a}), V(y^{\perp a} + y^{\| a})\rangle \\ &= \langle y^{\perp a}, Vy^{\perp a} \rangle \qquad\qquad\qquad\qquad (V^{1/2}y^{\| a} = 0) \\ &\leq \langle y^{\perp a}, \Sigma y^{\perp a} \rangle \qquad\qquad\qquad\qquad (V \preceq \Sigma), \end{aligned}$$

where the equality above can be attained by choosing $V$ so that the corresponding eigenvalues of $V$ along the direction of $y^{\perp a}$ coincide with those of $\Sigma$. Note that this is also feasible since the constraint

of eigenvalues being 0 only applies to the direction $y^{\|a}$, which is orthogonal to $y^{\perp a}$. To complete the proof, realize that the vector $y^{\perp a}$ could be constructed as follows:

$$y^{\perp a} = (I - a_0 a_0^T) y,$$

where $a_0 = a / \|a\|$ is the unit vector of $a$. The last step is to simplify the above equation as:

$$\langle y^{\perp a}, \Sigma y^{\perp a}, \rangle = \langle (I - a_0 a_0^T) y, \Sigma (I - a_0 a_0^T) y \rangle$$
$$= \operatorname{Var}(\mathbb{E}[Y \mid X]) - \left( \frac{2 \langle y, \Sigma a \rangle}{\langle a, a \rangle} - \frac{\operatorname{Var}(\mathbb{E}[A \mid X]) \cdot \langle a, y \rangle}{\langle a, a \rangle^2} \right) \langle a, y \rangle,$$

by using the fact that $\operatorname{Var}(\mathbb{E}[Y \mid X]) = \langle y, \Sigma y \rangle$ and $\operatorname{Var}(\mathbb{E}[A \mid X]) = \langle a, \Sigma a \rangle$.

To show when the equality is attained, let $V^*$ be the optimal solution of (**??**), which could be constructed by first eigendecomposing $\Sigma$ and then set all the eigenvalues of $\Sigma$ to 0 whose corresponding eigenvectors are not orthogonal to $a$. It is worth pointing out that $V^*$ is positive semidefinite but not necessarily invertible. Nevertheless, we could still define the projection matrix of $V^*$ that projects to the column space of $V^*$ as follows:

$$P_{V^*} := V^* (V^{*T} V^*)^\dagger V^{*T},$$

where $Q^\dagger$ denotes the Moore-Penrose pseudoinverse of matrix $Q$. With $P_{V^*}$, it is easy to verify that the optimal transformation is given by $Z$ such that

$$\mathbb{E}[\varphi(X) \mid Z] = P_{V^*} \varphi(X).$$

To see this, we have:

$$\operatorname{Cov}(\mathbb{E}[\varphi(X) \mid Z], \mathbb{E}[\varphi(X) \mid Z]) = \operatorname{Var} \mathbb{E}[\varphi(X) \mid Z]$$
$$= \operatorname{Var}(P_{V^*} \varphi(X))$$
$$= P_{V^*} \operatorname{Var}(\varphi(X)) P_{V^*}^T$$
$$= P_{V^*} \Sigma P_{V^*}^T$$
$$= V^*,$$

completing the proof. ∎

Next, we will prove Theorem 5.2, restated below:

**Theorem 5.2.** The optimal solution of optimization problem (12) is lower bounded by

$$\min_{Z, \operatorname{Var} \mathbb{E}[Y|Z] \geq \operatorname{Var}(Y)} \operatorname{Var} \mathbb{E}[A \mid Z] \geq \frac{\operatorname{Var}(Y) \cdot \langle a, y \rangle^2}{\langle y, y \rangle^2}. \tag{13}$$

The following theorem is the generalized version of Theorem 5.2 in noisy setting:

**Theorem C.2.** The optimal solution of optimization problem (12) is

$$\min_{Z, \operatorname{Var} \mathbb{E}[Y|Z] = \operatorname{Var}(Y)} \operatorname{Var} \mathbb{E}[A \mid Z] = \frac{\operatorname{Var}(\mathbb{E}[Y|X]) \cdot \langle a, y \rangle^2}{\langle y, y \rangle^2}. \tag{23}$$

It is easy to see Theorem 5.2 is an immediate corollary of this result: under the noiseless assumption, we have $\operatorname{Var}(\mathbb{E}[Y|X]) = \operatorname{Var}(Y)$ and $\operatorname{Var}(\mathbb{E}[A|X]) = \operatorname{Var}(A)$.

*Proof.* Due to the symmetry between $Y$ and $A$, here we only prove the first part of the theorem. As in the proof of Theorem 5.1, we have the following identities hold:

$$\operatorname{Var} \mathbb{E}[A \mid Z] = \langle a, \operatorname{Cov}(\mathbb{E}[\varphi(X) \mid Z], \mathbb{E}[\varphi(X) \mid Z]) a \rangle,$$
$$\operatorname{Var} \mathbb{E}[Y \mid Z] = \langle y, \operatorname{Cov}(\mathbb{E}[\varphi(X) \mid Z], \mathbb{E}[\varphi(X) \mid Z]) y \rangle.$$

Again, let $V := \operatorname{Cov}(\mathbb{E}[\varphi(X) \mid Z], \mathbb{E}[\varphi(X) \mid Z])$ so that we can relax the optimization problem as follows:

$$\min_Z \quad \langle a, V a \rangle, \qquad \text{subject to} \quad 0 \preceq V \preceq \Sigma, \langle y, V y \rangle = \operatorname{Var}(Y) = \langle y, \Sigma y \rangle.$$

To proceed, we first decompose $a$ orthogonally w.r.t. $y$:

$$a = a^{\perp y} + a^{\parallel y},$$

where $a^{\perp y}$ is the component of $a$ that is perpendicular to $y$ and $a^{\parallel y}$ is the parallel component of $a$ to $y$. Using this orthogonal decomposition, we have $\forall V$:

$$\langle a, Va \rangle = \langle (a^{\perp y} + a^{\parallel y}), V(a^{\perp y} + a^{\parallel y}) \rangle$$
$$\geq \langle a^{\parallel y}, Va^{\parallel y} \rangle, \qquad\qquad\qquad (V \succeq 0),$$

where the equality could be attained by choosing $V$ such that $V^{1/2}a^{\perp y} = 0$. On the other hand, it is clear that

$$a^{\parallel y} = \langle a, y_0 \rangle \cdot y_0,$$

where $y_0 = y/\|y\|$ is the unit vector of $y$. Plug $a^{\parallel y} = \langle a, y_0 \rangle \cdot y_0$ into $\langle a^{\parallel y}, Va^{\parallel y} \rangle$ with the fact that $\langle y, Vy \rangle = \mathrm{Var}(\mathbb{E}[Y|X]) = \langle y, \Sigma y \rangle$, we get

$$\langle a^{\parallel y}, Va^{\parallel y} \rangle = \frac{\mathrm{Var}(\mathbb{E}[Y|X]) \cdot \langle a, y \rangle^2}{\langle y, y \rangle^2}.$$

Again, to attain the equality, we should first construct the optimal $V^*$ matrix by eigendecomposing $\Sigma$. Specifically, this time we set all the eigenvalues of $\Sigma$ whose corresponding eigenvectors are perpendicular to $y$ to 0. Similar to what we argue in the proof of Theorem 5.1, $V^*$ is positive semidefinite but not necessarily invertible. Nevertheless, we could still define the projection matrix of $V^*$ that projects to the column space of $V^*$ as follows:

$$P_{V^*} := V^*(V^{*T}V^*)^\dagger V^{*T},$$

where $Q^\dagger$ denotes the Moore-Penrose pseudoinverse of matrix $Q$. With $P_{V^*}$, it is easy to verify that the optimal transformation is given by $Z$ such that

$$\mathbb{E}[\varphi(X) \mid Z] = P_{V^*}\varphi(X).$$

To see this, we have:

$$\mathrm{Cov}(\mathbb{E}[\varphi(X) \mid Z], \mathbb{E}[\varphi(X) \mid Z]) = \mathrm{Var}\,\mathbb{E}[\varphi(X) \mid Z]$$
$$= \mathrm{Var}(P_{V^*}\varphi(X))$$
$$= P_{V^*}\mathrm{Var}(\varphi(X))P_{V^*}^T$$
$$= P_{V^*}\Sigma P_{V^*}^T$$
$$= V^*,$$

completing the proof. ∎

## C.3    Proof of Theorem 5.3

To prove Theorem 5.3, we first introduce the following decompositions of the loss functions:

The following lemma is a more refined version of the Data-Processing Inequality, which gives an exact characterization of the Bayes optimality gap for a given $Z$. Recall that the Bayes error is $\mathbb{E}_X[\mathrm{Var}[Y|X]]$.

**Lemma C.1** ($L^2$ Error Decomposition)**.**

$$\mathbb{E}_Z[\mathrm{Var}[Y|Z]] - \mathbb{E}_X[\mathrm{Var}[Y|X]] = \mathbb{E}_Z\,\mathrm{Var}\,(\mathbb{E}[Y|X]\,|Z) \geq 0. \qquad (24)$$

Similarly,

$$\mathbb{E}_Z[\mathrm{Var}[A|Z]] - \mathbb{E}_X[\mathrm{Var}[A|X]] = \mathbb{E}_Z\,\mathrm{Var}\,(\mathbb{E}[A|X]\,|Z) \geq 0. \qquad (25)$$

*Proof.* Since $Z = g(X)$ is a function of $X$, we have $p(y|x) = p(y|x,z)$, or equivalently, $(Y \perp Z)|X$

By law of total variance,

$$\mathrm{Var}(Y|Z) = \mathbb{E}_X\left[\mathrm{Var}(Y|X,Z)|Z\right] + \mathrm{Var}\left(\mathbb{E}[Y|X,Z]\,|Z\right)$$
$$= \mathbb{E}_X\left[\mathrm{Var}(Y|X)|Z\right] + \mathrm{Var}\left(\mathbb{E}[Y|X]\,|Z\right)$$

Taking expectation over $Z$,

$$\mathbb{E}_Z \operatorname{Var}(Y|Z)$$
$$= \mathbb{E}_Z \mathbb{E}_X \left[ \operatorname{Var}(Y|X)|Z \right] + \mathbb{E}_Z \operatorname{Var}\left( \mathbb{E}\left[Y|X\right]|Z \right)$$
$$= \mathbb{E}_X \mathbb{E}_Z \left[ \operatorname{Var}(Y|X)|Z \right] + \mathbb{E}_Z \operatorname{Var}\left( \mathbb{E}\left[Y|X\right]|Z \right)$$
$$= \mathbb{E}_X \operatorname{Var}(Y|X) + \mathbb{E}_Z \operatorname{Var}\left( \mathbb{E}\left[Y|X\right]|Z \right),$$

where the last equality is due to the law of total expectation. ∎

The following lemma is a direct consequence of the law of total variance.

**Lemma C.2** ($L^2$ Invariance Decomposition)**.**

$$\operatorname{Var}(A) - \mathbb{E}_Z \operatorname{Var}(A|Z) = \operatorname{Var}(\mathbb{E}[A|Z]) \geq 0.$$

We will prove a generalized version of Theorem 5.3 without noiseless assumption, stated below:

**Theorem C.3.** The optimal solution of the Lagrangian has the following lower bound:

$$\textbf{OPT}(\lambda) \geq \frac{1}{2} \Big\{ \lambda \operatorname{Var}(\mathbb{E}[A|X]) + \operatorname{Var}(\mathbb{E}[Y|X]) - \sqrt{(\lambda \operatorname{Var}(\mathbb{E}[A|X]) + \operatorname{Var}(\mathbb{E}[Y|X]))^2 - 4\lambda\langle a, \Sigma y\rangle^2} \Big\}$$
$$+ (\mathbb{E}[\operatorname{Var}(Y|X)] - \lambda \operatorname{Var}(A)).$$

When the noiseless assumption holds, we have $\operatorname{Var}(\mathbb{E}[A|X]) = \operatorname{Var}(A)$, $\operatorname{Var}(\mathbb{E}[Y|X]) = \operatorname{Var}(Y)$, and $\mathbb{E}[\operatorname{Var}(Y|X)] = 0$, hence the bound above simplifies to:

$$\frac{1}{2} \Big\{ \operatorname{Var}(Y) - \lambda \operatorname{Var}(A) - \sqrt{(\lambda \operatorname{Var}(A) + \operatorname{Var}(Y))^2 - 4\lambda\langle a, \Sigma y\rangle^2} \Big\}.$$

which is exactly Theorem 5.3.

*Proof of Theorem 5.3.* By Lemma C.1 and Lemma C.2, we can decompose the objective as:

$$\mathbb{E}[\operatorname{Var}(Y|Z)] - \lambda \mathbb{E}[\operatorname{Var}(A|Z)]$$
$$= (\mathbb{E}[\operatorname{Var}(Y|Z)] - \mathbb{E}[\operatorname{Var}(Y|X)]) + \lambda(\operatorname{Var}(A) - \mathbb{E}[\operatorname{Var}(A|Z)]) + (\mathbb{E}[\operatorname{Var}(Y|X)] - \lambda \operatorname{Var}(A))$$
$$= \mathbb{E}_Z \operatorname{Var}\left( \mathbb{E}\left[Y|X\right]|Z \right) + \operatorname{Var}(\mathbb{E}[A|Z]) + (\mathbb{E}[\operatorname{Var}(Y|X)] - \lambda \operatorname{Var}(A))$$

Since $\mathbb{E}[\operatorname{Var}(Y|X)] - \lambda \operatorname{Var}(A)$ does not depend on $Z$, we will focus on the first two terms:

$$\min_{Z=g(X)} \quad \mathbb{E}_Z \operatorname{Var}\left( \mathbb{E}\left[Y|X\right]|Z \right) + \lambda \operatorname{Var}(\mathbb{E}[A|Z]). \tag{26}$$

Recall that for the squared loss,

$$f_Y^*(X) = \mathbb{E}\left[Y|X\right], \quad f_A^*(X) = \mathbb{E}\left[A|X\right].$$

We will first simplify the objective in (26). We have

$$\mathbb{E} \operatorname{Var}\left( \mathbb{E}\left[Y|X\right]|Z \right) = \mathbb{E} \operatorname{Var}\left( f_Y^*(X)|Z \right), \tag{27}$$

and using the law of total expectation,

$$\mathbb{E}(A|Z) = \int \mathbb{E}(A|X, Z) p(X|Z) dX.$$

Since $Z = g(X)$ is a function of $X$, we have $Z \perp A|X$, so $\mathbb{E}(A|X, Z) = \mathbb{E}(A|X) = f_A^*(X)$. Therefore,

$$\mathbb{E}[A|Z] = \int \mathbb{E}[A|X, Z] p(X|Z) dX$$
$$= \int f_A^*(X) p(X|Z) dX = \mathbb{E}[f_A^*(X)|Z]$$

Hence,

$$\operatorname{Var}(\mathbb{E}(A|Z)) = \operatorname{Var}(\mathbb{E}[f_A^*(X)|Z]) \tag{28}$$

Now we substitute (27), (28) into (26), which gives the following equivalent form of (26):

$$\min_{Z=g(X)} \left\{ \mathbb{E}\operatorname{Var}\left(f_Y^*(X)|Z\right) + \lambda\operatorname{Var}(\mathbb{E}[f_A^*(X)|Z]) \right\}$$

In this case, the objective (29) becomes:

$$\mathbb{E}\operatorname{Var}\left(f_Y^*(X)|Z\right) + \lambda\operatorname{Var}(\mathbb{E}[f_A^*(X)|Z]) \tag{29}$$

$$=\mathbb{E}\operatorname{Var}\left(\langle y, \varphi(X)\rangle|Z\right) + \lambda\operatorname{Var}(\mathbb{E}[\langle a, \varphi(X)\rangle|Z]) \tag{30}$$

$$=\langle y, \mathbb{E}\operatorname{Cov}(\varphi(X), \varphi(X)|Z)y\rangle + \tag{31}$$

$$\lambda\langle a, \operatorname{Cov}(\mathbb{E}[\varphi(X)|Z], \mathbb{E}[\varphi(X)|Z])a\rangle \tag{32}$$

By the law of total covariance,

$$\mathbb{E}\operatorname{Cov}(\varphi(X), \varphi(X)|Z) + \operatorname{Cov}(\mathbb{E}[\varphi(X)|Z], \mathbb{E}[\varphi(X)|Z])$$
$$= \operatorname{Cov}(\varphi(X), \varphi(X)) = \Sigma$$

Let $V = \operatorname{Cov}(\mathbb{E}[\varphi(X)|Z], \mathbb{E}[\varphi(X)|Z])$ , which satisfies $\Sigma \succeq V \succeq 0$. Then, finding the feature transform $Z = g(X)$ that minimizes (32) is equivalent to:

$$\min_{V=\operatorname{Cov}(\mathbb{E}[\varphi(X)|Z], \mathbb{E}[\varphi(X)|Z])} \langle y, (\Sigma - V)y\rangle + \lambda\langle a, Va\rangle$$

The key technique of our lower bound is to relax the constraint $V = \operatorname{Cov}(\mathbb{E}[\varphi(X)|Z], \mathbb{E}[\varphi(X)|Z])$ by the semi-definite constraint $\Sigma \succeq V \succeq 0$.

$$\min_{V:\Sigma\succeq V\succeq 0} \langle y, (\Sigma - V)y\rangle + \lambda\langle a, Va\rangle$$

This is an SDP whose optimal solution lower bounds the objective (26). Moreover, we can show that there is a simplified form for the SDP optimal solution using eigenvalues and eigenvectors:

$$\langle y, (\Sigma - V)y\rangle + \lambda\langle a, Va\rangle$$
$$=\langle y, \Sigma y\rangle + \langle V, \lambda aa^T - yy^T\rangle$$
$$=\langle y, \Sigma y\rangle + \langle \Sigma^{-1/2}V\Sigma^{-1/2}, \Sigma^{1/2}(\lambda aa^T - yy^T)\Sigma^{1/2}\rangle.$$

Note that $I \succeq Q := \Sigma^{-1/2}V\Sigma^{-1/2} \succeq 0$, and $R := \Sigma^{1/2}(\lambda aa^T - yy^T)\Sigma^{1/2}$ is a matrix with rank at most 2.

When the matrix $R$ is positive definite or negative definite, the minimum is achieved at $Q = 0$ or $I$. Otherwise, the only possibility is that $R$ is a rank-2 matrix with one positive eigenvalue and one negative eigenvalue. By Von-Neumann's trace inequality,

$$\langle Q, R\rangle \geq \sum_{i=1}^{d} \sigma_i(R)\sigma_{d-i+1}(Q).$$

Since $\sigma_1(R) > 0 = \sigma_2(R) = ... = \sigma_{d-1}(R) = 0 > \sigma_d(R)$ and $0 \leq \sigma_d(Q) \leq 1$, we have

$$\langle Q, R\rangle \geq \sigma_d(R) = \sigma_d(\Sigma^{1/2}(\lambda aa^T - yy^T)\Sigma^{1/2})$$

The minimizer is

$$Q = ww^T, V = \Sigma^{1/2}ww^T\Sigma^{1/2}$$

where $w$ is the unit eigenvector of $R$ with eigenvalue $\sigma_d(R)$. By Lemma C.3,

$$\sigma_d(R) = \frac{1}{2}\left\{ \lambda\langle a, \Sigma a\rangle - \langle y, \Sigma y\rangle - \sqrt{(\lambda\langle a, \Sigma a\rangle + \langle y, \Sigma y\rangle)^2 - 4\lambda\langle a, \Sigma y\rangle^2} \right\}$$

Therefore,

$$\mathbf{OPT}(\lambda) = \langle y, (\Sigma - V)y\rangle + \lambda\langle a, Va\rangle + (\mathbb{E}[\operatorname{Var}(Y|X)] - \lambda\operatorname{Var}(A))$$
$$\geq \langle y, \Sigma y\rangle + \sigma_d(R) + (\mathbb{E}[\operatorname{Var}(Y|X)] - \lambda\operatorname{Var}(A))$$
$$\geq \frac{1}{2}\left\{ \lambda\langle a, \Sigma a\rangle + \langle y, \Sigma y\rangle - \sqrt{(\lambda\langle a, \Sigma a\rangle + \langle y, \Sigma y\rangle)^2 - 4\lambda\langle a, \Sigma y\rangle^2} \right\}$$
$$+ (\mathbb{E}[\operatorname{Var}(Y|X)] - \lambda\operatorname{Var}(A))$$
$$= \frac{1}{2}\left\{ \lambda\operatorname{Var}(\mathbb{E}[A|X]) + \operatorname{Var}(\mathbb{E}[Y|X]) - \sqrt{(\lambda\operatorname{Var}(\mathbb{E}[A|X]) + \operatorname{Var}(\mathbb{E}[Y|X]))^2 - 4\lambda\langle a, \Sigma y\rangle^2} \right\}$$
$$+ (\mathbb{E}[\operatorname{Var}(Y|X)] - \lambda\operatorname{Var}(A))$$

Hence we have completed the proof. ∎

## C.4  EXPLICIT FORMULA FOR EIGENVALUES

The following lemma is used the in the last step of the proof of Theorem 5.3 to simplify the expression involving $\sigma_d(R) = \sigma_d(\Sigma^{1/2}(\lambda aa^T - yy^T)\Sigma^{1/2})$.

**Lemma C.3.** Let $R = \Sigma^{1/2}(\lambda aa^T - yy^T)\Sigma^{1/2}$, then the eigenvalues of $R$ are

$$\sigma_1(R) = \frac{1}{2}\Big\{\lambda\langle a, \Sigma a\rangle - \langle y, \Sigma y\rangle + \sqrt{(\lambda\langle a, \Sigma a\rangle + \langle y, \Sigma y\rangle)^2 - 4\lambda\langle a, \Sigma y\rangle^2}\Big\},$$

$$\sigma_d(R) = \frac{1}{2}\Big\{\lambda\langle a, \Sigma a\rangle - \langle y, \Sigma y\rangle - \sqrt{(\lambda\langle a, \Sigma a\rangle + \langle y, \Sigma y\rangle)^2 - 4\lambda\langle a, \Sigma y\rangle^2}\Big\}$$

$$\sigma_2(R) = \cdots = \sigma_{d-1}(R) = 0$$

*Proof.* Since $\text{rank}(R) \leq \text{rank}(\lambda aa^T - yy^T) \leq 2$, $R$ has at most two non-zero eigenvalues $\sigma_1(R)$ and $\sigma_d(R)$. Notice that

$$\text{tr}(R) = \sum_{i=1}^d \sigma_i(R) = \sigma_1(R) + \sigma_d(R),$$

$$\text{tr}(R^2) = \sum_{i=1}^d \sigma_i^2(R) = \sigma_1^2(R) + \sigma_d^2(R)$$

We can write $\text{tr}(R)$ and $\text{tr}(R^2)$ explicitly:

$$\text{tr}(R) = \lambda\,\text{tr}(\Sigma^{1/2}aa^T\Sigma^{1/2}) - \text{tr}(\Sigma^{1/2}yy^T\Sigma^{1/2}) = \lambda\langle a, \Sigma a\rangle - \langle y, \Sigma y\rangle$$

$$\text{tr}(R^2) = \text{tr}(\Sigma^{1/2}(\lambda aa^T - yy^T)\Sigma(\lambda aa^T - yy^T)\Sigma^{1/2})$$
$$= \text{tr}(\Sigma^{1/2}\left(\lambda^2(a^T\Sigma a)aa^T - \lambda a^T\Sigma y(a\Sigma y^T) - \lambda y^T\Sigma a(y\Sigma a^T) + (a^T\Sigma a)aa^T\right)\Sigma^{1/2})$$
$$= \lambda^2(a^T\Sigma a)^2 - 2\lambda(a^T\Sigma y)^2 + (y^T\Sigma y)^2$$

Therefore,

$$\sigma_1(R) + \sigma_d(R) = \lambda\langle a, \Sigma a\rangle - \langle y, \Sigma y\rangle$$

$$\sigma_1(R)\sigma_d(R) = \frac{1}{2}\left((\sigma_1(R) + \sigma_d(R))^2 - (\sigma_1^2(R) + \sigma_d^2(R))\right)$$
$$= \lambda\langle a, \Sigma y\rangle^2 - \lambda\langle a, \Sigma a\rangle\langle y, \Sigma y\rangle$$

Thus $\sigma_1(R)$ and $\sigma_d(R)$ are the roots of the quadratic equation:

$$x^2 - (\lambda\langle a, \Sigma a\rangle - \langle y, \Sigma y\rangle)x + \lambda\langle a, \Sigma y\rangle^2 - \lambda\langle a, \Sigma a\rangle\langle y, \Sigma y\rangle = 0$$

We complete the proof by solving this quadratic equation. ∎

## C.5  ACHIEVABILITY OF LOWER BOUND

In the proof of Theorem 5.3, we showed a lower bound on the tradeoff via an SDP relaxation. Therefore, the lower bound is achievable whenever the SDP relaxation is tight. We state this as a regularity condition on $(X, \varphi)$.

**Definition C.1.** $(X, \varphi)$ is called **regular**, if for any positive semidefinite matrix $M$: $\Sigma \succeq M \succeq 0$, there exists $Z = g(X)$, such that

$$\text{Cov}(\mathbb{E}[\varphi(X)|Z], \mathbb{E}[\varphi(X)|Z]) = M.$$

**Theorem C.4.** When $(X, \varphi)$ is regular, the lower bound in Theorem 5.3 is achievable.

*Proof.* From the proof of Theorem 5.3, we can see that if there exists $Z = g(X)$, such that

$$\text{Cov}(\mathbb{E}[\varphi(X)|Z], \mathbb{E}[\varphi(X)|Z]) = \Sigma^{1/2}ww^T\Sigma^{1/2},$$

where $w$ is the unit eigenvector of $R$ with eigenvalue $\sigma_d(R)$, then the equality is achievable. It is easy to see that

$$\Sigma \succeq \Sigma^{1/2}ww^T\Sigma^{1/2} \succeq 0.$$

Therefore, choosing $M = \Sigma^{1/2}ww^T\Sigma^{1/2}$ in the definition of regularity guarantees the existence of $Z$. Hence we have completed the proof. ∎

A sufficient condition on the regularity of $(X, \varphi)$ is the Gaussianity of $\varphi(X)$, in which case choosing $g(X)$ as a linear transform is sufficient:

**Theorem C.5.** $(X, \varphi)$ is regular if $\varphi(X)$ follows Gaussian distribution.

*Proof.* Note that when $\varphi(X)$ is Gaussian, $(\varphi(X), L\varphi(X))$ is jointly Gaussian for any $L \in \mathbb{R}^{k \times d}$. Let $Z = L\varphi(X)$, then the conditional distribution $\varphi(X)|Z$ is Gaussian, with mean and covariance

$$\mathbb{E}[\varphi(X)|Z] = \Sigma L^T (L\Sigma L^T)^{-1} Z, \operatorname{Cov}[\varphi(X), \varphi(X)|Z]$$
$$= \Sigma L^T (L\Sigma L^T)^{-1} L\Sigma.$$

Hence,

$$\mathbb{E}\operatorname{Cov}[\varphi(X), \varphi(X)|Z] = \Sigma L^T (L\Sigma L^T)^{-1} L\Sigma.$$

We will prove that for any $\Sigma \succeq M \succeq 0$, there exists a linear transform $L$, such that $M = \Sigma L^T (L\Sigma L^T)^{-1} L\Sigma$.

Consider the eigenvalue decomposition of $\Sigma^{-1/2} M \Sigma^{-1/2} = U^T D U$, $k = rank(M), U \in \mathbb{R}^{k \times d}, D \in \mathbb{R}^{k \times k}$, $D$ is invertible. Then, let $L = D^{-1/2} U \Sigma^{-1/2}$, we have

$$L\Sigma L^T = D^{-1},$$
$$L\Sigma = D^{-1/2} U \Sigma^{1/2},$$
$$\Sigma L^T (L\Sigma L^T)^{-1} L\Sigma = M.$$

Therefore we have completed the proof. ∎

We conjecture that this regularity condition holds for more general distributions beyond Gaussian.

