# OpenReview forum: "Fundamental Limits and Tradeoffs in Invariant Representation Learning"
_ICLR.cc/2021/Conference — Reject_

### Official Review · AnonReviewer3 · 2020-10-28
**Very promising approach!**

**Rating:** 5
**Confidence:** 1

**Review:**

Authors extend the information bottleneck by Tishby to also include an additional auxiliary variable that represents either domain index or private information. Using this generalined IB the authors study tradeoffs/fundamental limtis on achievalbe accuracy in predicting labels while maintaining invariance against the auxialiary variable.

The proposed approach seems promising but much more work is needed:
- it is unclear how the proposed frameworl relates precisely to plain vanilla IB. I feeld that convential IB could be obtained somehow when using the auxiliary variable to identify individual traning samples instead of larger domains.

- how can the obtained characterization be used ? can one use these characterizations to verify sub-optimiality of existing methods for privacy-preserving ML or algorithmic fairness

- how can the proposed model be turned into practical algorithms that (nearly) achieve the fundamental limits/tradeoffs.

minor issues:
- pls indicate the optimization domain for every optimization problem (e.g. (1))
- unclear what is meant by "could be the identity of domain index in domain adaptation,"
- "..Lagrangian has the following lower bound:.."
- "Evidently, the key quantity in the lower bound..." why is this evident ?

---

> ### Author Response · Authors · 2020-11-19
> **Response**
>
> We’d like to thank the reviewer for the comments, and we want to take this change to clarify the misconception in the review. As we pointed out in Example 2.5, our framework is fundamentally different from the IB framework, due to the additional variable.
>
> Q: “the conventional IB could be obtained...”
>
> A: Note that as we mentioned in Example 2.5, the IB formulation is $I(Y; Z) - \lambda I(X; Z)$, and its goal is to maximize predictive accuracy while compressing the representations as much as possible. As a comparison, we don’t have any constraint on the compression of data, but rather an invariant constraint.
>
> Q: “can one use … to verify sub-optimality?”
>
> A: Yes, and we discussed this on page 7 before Section 5. For example, in algorithmic fairness, given a fair representation, we could compute the corresponding information-theoretic quantities, e.g., $I(A; Z)$ and $I(Y; Z)$, and use its distance to the line segment as a certificate of the non-optimality of this representation.
>
> Q: “how can the proposed model be turned into practical algorithms that (nearly) achieve the fundamental limits”
>
> A: For g(X) that achieves the extremal point, e.g., the optimal one, our proof specifies how to construct such g(X) in certain cases (e.g., at the end of page 13 in appendix). But even in these cases, the construction of the optimal g(X) requires knowledge about the underlying population distribution, which is often not available in learning. From this perspective, we view our work as a pioneer one that inspires future algorithmic contributions towards this important open problem.
>
> Q: “domain index in domain adaptation”
>
> A: Here we mean that the value of A corresponds to the identifier of each domain. For example A = 0 means it’s the source domain and A = 1 means it’s the target domain, etc.
>
> Q: Other minor issues:
>
> A: We will be sure to incorporate these to update our paper.

---

### Official Review · AnonReviewer1 · 2020-11-01
**Studies a very important problem in representation learning but the implication is vague**

**Rating:** 5
**Confidence:** 3

**Review:**

This work studies a fundamental and important problem in representation learning, the tradeoff between accuracy and invariance of the learned representations. For classification and regression problems, the paper analyzed the inherent tradeoffs by providing a geometric characterization of the feasible region in the information plane, where it connects the geometric properties of this feasible region to the fundamental limitations of the tradeoff problem. The work is well written and well presented overall, but the implication of the result is not very clear and the work lacks experimental validations.
Below are some major comments/concerns from the reviewer:

1. One of the major questions the reviewer has is what is the implication of the result. In the other words, given the characterization of the feasible region in the information plane, including its boundedness, convexity, and extremal vertices, how can the result in this paper improve representation learning? What is the right balance of classification and invariance? It seems quite vague in the current form of the work.

2. Does Assumption 4.1 and Assumption 5.1 hold in general? Does the mapping f always exist, or under what condition does the mapping f exist? More discussion about the assumptions is needed. It is not how stringent the assumptions are in practice.

3. Another question the paper does not address, is how to construct g(X) to achieve certain points on the information plane.

4. Although the result is theoretical, it could be better if the authors provide some experimental justifications, which is completely missing.

---

> ### Author Response · Authors · 2020-11-19
> **Response**
>
> We’d like to thank the reviewer for the comments.
> Q: “what is the implication of the result”
>
> A:  One direct implication of the convexity is that we could improve existing deterministic representation learning methods by allowing randomized representations. To be more specific, one immediate corollary of our result is that, given two representations, we could construct a new one by randomizing between these two to achieve a new accuracy-invariance tradeoff, and this would allow us to interpolate between existing invariances and accuracies.
>
> The extremal vertices also specify the limit of accuracy/invariance under the constraint of the other. As the reviewer has mentioned, this is a fundamental and important problem in representation learning.
>
> Q: Assumption about 4.1 and 5.1
>
> A: Assumption 4.1 and 5.1 are standard assumptions in the original PAC learning theory (https://en.wikipedia.org/wiki/Probably_approximately_correct_learning), and they correspond to the notion of “concept” that a learner aims to recover. These correspond to the noiseless setting. In appendix we also provide proof for the general noisy setting.
>
> Q: “how to construct g(X)”
>
> A: As we explained above, for g(X) whose corresponding point is in the interior of the feasible region, i.e., non-optimal representations, such g(X) could be constructed using randomization, provided that two existing representations are provided. For g(X) that achieves the extremal point, e.g., the optimal one, our proof also specifies how to construct such g(X) in certain cases (e.g., at the end of page 13 in appendix). But even in these cases, the construction of the optimal g(X) requires knowledge about the underlying population distribution, which is often not available in learning.
>
> Q: “could be better if … experimental justifications”
>
> A: We’d love to, but we are not clear what kind of experimental results could better justify our theoretical results than the proof itself. For example, there is no existing algorithm that provably achieves the optimal extremal points.

---

### Official Review · AnonReviewer5 · 2020-11-08
**A geometric approach to invariant representations**

**Rating:** 5
**Confidence:** 2

**Review:**

The paper formulates the problems of learning in invariant representations as a min-max game, exploring tradeoffs between accuracy and invariance of these representations via a geometric plane analysis. Specifically. the paper considers both classification (cross entropy loss) and regressions settings (squared loss).The related minimax problem is separable in the sense that for any fixed feature transformation, the optimization for the min and max agent are independent of each other, resulting is a simple, concise representation of the resulting optimization problem.  The symmetric nature of this description allows for a geometric description of the feasible set in regards to the actions of both min-max agents and the paper goes on to provide some characterizations of external points and other properties (e.g. convexity) of this region/set. The paper also derives a tight lower bound that for the Lagrangian form of accuracy and invariance.

This is an interesting theoretical take on the issue of exploring tradeoffs between accuracy and invariance of representations, but I am not sure to what extend this theoretical analysis provides actionable insights about real world problems (e.g. what doe convexity of the feasible set imply for algorithmic fairness?). The paper would be much more convincing if it offered some concrete use of these ideas in examples/applications.

Along these lines, I am not sure about whether some modelling assumptions are always satisfied in practice. For example, it seems to me that in the analysis both Y/A tasks are considered to be either jointly classification or jointly regression tasks but in many of the examples put forward it could be the Y is a continuous variable e.g. salary where A could correspond to a discrete one e.g. gender. What does this framework imply in this case?  How sensitive is it on the choice of the loss functions?

Overall, this geometric approach seems like a cute idea but at least in its current form maybe it is not overly insightful for the concrete applications that it aims to model.

---

> ### Author Response · Authors · 2020-11-19
> **Response**
>
> We’d like to thank the reviewer for the comments.
>
> Q: “to what extent … actionable insights about the real-world problems”
>
> A:  As we briefly mentioned before Section 5, for the aforementioned applications, our characterization of the frontier is critical in order to be able to certify that a given model is not optimal. For example, in algorithmic fairness, given a fair representation, we could compute the corresponding information-theoretic quantities, e.g., $I(A; Z)$ and $I(Y; Z)$, and use its distance to the line segment as a certificate of the non-optimality of this representation. We would also like to point out that our proof of the convexity is constructive. So for example in algorithmic fairness, given two fair representations, the convexity shows that we could construct a randomized fair representation from the given two that achieves a compromise between fairness and accuracy.
>
>
> Q: Mixed discrete and continuous Y/A
>
> A: Our current analysis cannot be straightforwardly extended to this setting. It’s beyond the scope of this work hence left as future work. That being said, on empirical data, one could discretize the continuous variable and apply our classification setting since only a finite number of values could be taken in an empirical distribution.

---

> > ### Comment · AnonReviewer5 · 2020-11-21
> > **Response**
> >
> > Thank you for your response. Although at a very high level the ideas about exploring tradeoffs between fairness and accuracy sound promising, I think the paper would be much stronger if it explored such connections more thoroughly. This is not just about the issue of showing that a theoretical abstraction is useful. I believe that such a grounding would help by making some of the concepts in the paper more concrete and easier to parse.

---

> > > ### Author Response · Authors · 2020-11-21
> > > **Response: Thanks for the update, but our contributions are not just limited to algorithmic fairness**
> > >
> > > Thank you for the updated comment. We acknowledge that the concepts and results in the manuscript are a bit abstract, but we also believe this level of abstraction from specific applications makes our results more widely applicable in broader domains, including but not limited to fairness, domain adaptation, privacy, etc. Our focus of this paper is not to emphasize a particular application, but rather provide a unified analysis so that it could be applied to many relevant applications listed in Section 2. Furthermore, we would like to bring it to the reviewer's attention that we do provide a discussion on the interpretation of our results in Section 3.

---

### Author Response · Authors · 2020-11-19
**General Response**

We thank all the reviewers for the feedback, and we appreciate that all the reviewers acknowledge the fundamental importance of the problem we study in this work. It seems that the main concern is about the implication of our theoretical results, which we have addressed separately in our response to each of the reviewers. We look forward to your feedback!

---

### Decision · Program_Chairs · 2021-01-07
**Final Decision**

**Decision:**

Reject

**Comment:**

This paper studies an interesting information-theoretic trade-off between accuracy and invariance by posing it as a minimax problem. The results are of theoretical nature. However, the implications of the results are not clear. Also, the model/assumptions authors consider are not completely justified. Therefore, the paper at this stage is not recommended for acceptance. However, I highly encourage the authors to improve upon their existing work and resubmit to the next ML conference.